# Nutritional Iodine Status in Pregnant Women from Health Area IV in Asturias (Spain): Iodised Salt Is Enough

**DOI:** 10.3390/nu13061816

**Published:** 2021-05-27

**Authors:** Silvia González-Martínez, María Riestra-Fernández, Eduardo Martínez-Morillo, Noelia Avello-Llano, Elías Delgado-Álvarez, Edelmiro Luis Menéndez-Torre

**Affiliations:** 1Endocrinology and Nutrition Service, Hospital Universitario Central de Asturias (HUCA), 33011 Oviedo, Spain; eliasdelga@gmail.com (E.D.-Á.); edelangot@gmail.com (E.L.M.-T.); 2Endocrinology, Nutrition, Diabetes and Obesity Research Group (ENDO), Health Research Institute of Principado de Asturias (ISPA), 33011 Oviedo, Spain; mriestra.fernandez@gmail.com; 3Endocrinology and Nutrition Service, Hospital Universitario de Cabueñes, 33203 Gijón, Spain; 4Clinical Biochemistry Service, Hospital Universitario Central de Asturias (HUCA), 33011 Oviedo, Spain; edumartinezmorillo@gmail.com (E.M.-M.); noelia.avello@sespa.es (N.A.-L.); 5Faculty of Medicine, University of Oviedo, 33011 Oviedo, Spain

**Keywords:** iodine, iodine nutrition state, pregnancy, urinary iodine concentration, thyroid function, iodised salt, iodine supplements

## Abstract

Background: Iodine deficiency during pregnancy may have adverse effects on the neurodevelopment of the foetus. Recent studies of pregnant women in Asturias (Spain) indicate that nutritional iodine levels are sufficient. The objective of this study was to confirm the appropriate nutritional iodine status and to analyse the influence of the ingestion of iodine on maternal urinary iodine concentration (UIC) and thyroid function. Methods: An observational study was carried out between May and June 2017 on women in the first trimester of pregnancy from Health Area IV in Asturias. The women completed a questionnaire related to their consumption of iodine and samples were taken to analyse UIC and thyroid function. Results: Three hundred and eighteen pregnant women were involved. Of these, 51.10% used iodised salt, 48.90% consumed ≥ 2 servings of dairy products daily and 87.08% took iodine supplements. The median UIC was 171.5 μg/L (116–265 μg/L) and 60.41% of women had UIC ≥ 150 μg/L. Multivariate logistic regression analysis demonstrated that iodised salt had a protective effect on UIC < 150 μg/L (odds ratio (OR) 0.404 (0.237–0.683), *p* = 0.001), but not iodine supplements (OR 0.512 (0.240–1.085), *p* = 0.080). The average level of thyroid stimulating hormone (TSH) was 2.26 ± 0.94 mIU/L; 68.40% of pregnant women taking iodine supplements had TSH < 2.5 mIU/L compared to 30.00% of those who were not taking supplements (*p* = 0.031). Conclusions: The pregnant women in our health area are maintaining appropriate nutritional iodine levels. The consumption of iodised salt protects against iodine deficiency; thus, iodine supplements should be taken on an individualised basis.

## 1. Introduction

Micronutrient iodine is essential for the correct functioning of the thyroid gland. Thyroid hormones are responsible for a range of functions, from basal metabolism and heart rate to bone growth and development of the central nervous system [1,2]. During gestation, severe iodine deficiency is related to both maternal and foetal hypothyroidism as well as serious neurological defects in the child [3,4,5]. Based on observational studies, it has also been suggested that slight to moderate deficit, even if normal maternal thyroid function is maintained, may be associated with neurodevelopmental alterations in the child, which range from reduced IQ scores to varying levels of impairment in executive functioning and language or reading abilities [6,7,8,9].

The World Health Organization (WHO) estimated in 2007 that two billion people have insufficient iodine intake [10], making it a large-scale public health problem and the primary cause of preventable neurological damage. The principal strategy for tackling iodine deficit is iodine fortification of table salt [11], a measure that has been shown to be successful and cost-effective [12].

The most commonly used method to establish the iodine nutrition situation in the general population is measuring urinary iodine concentration (UIC), or ioduria, in the school population [13]. However, results from school children cannot be extrapolated to pregnant women. This is because of the higher iodine requirements during pregnancy due to increased renal clearance, the de-iodinising action on the placenta, increased maternal thyroxin synthesis and transfer of iodine to the foetus [14,15,16]. For this reason, pregnant women should increase their daily ingestion of iodine to 250 μg [17]. Thus, to ensure this population has sufficient iodine levels, various national and international scientific entities recommend universal pharmacological iodine supplements for pregnant women [18,19,20].

In 2007, however, the WHO published a controversial consensus document that indicated that for pregnant women in countries or regions with adequate iodine ingestion, defined as a median UIC > 100 μg/L for more than 2 years and the use of iodised salt in >90% of homes, there was no need for supplementation: pregnant women were protected simply by the ingestion of iodised salt [21].

Since 2004, Spain has been among the countries where nutritional iodine status is considered sufficient in the general population. Studies of pregnant women, however, demonstrate that there are disparities between the different autonomous regions and an increased prevalence of iodine deficiency [22,23,24].

Despite this finding, in 2012 a workshop was carried out in Spain on “Iodine and folic acid supplementation during pregnancy and breastfeeding” [25]. One of the final recommendations of the workshop was that if the ingestion of iodine through iodised salt and dairy products was adequate “universal supplementation with potassium iodide during pregnancy and breastfeeding is not justified currently in Spain”.

In Asturias (Spain), in 1982 a Public Health Campaign was put in place to promote the use of iodised salt. Thanks to this, studies are available that demonstrate that optimal iodine nutrition has been the norm since 1992, and as a result, the eradication of iodine deficiency in the region was announced in 2000 [26]. The most recent review of nutritional iodine status was conducted in 2010; the results show the median UIC of the school population to be 180.7 μg/L and that in women of child-bearing age to be 170.6 μg/L [27].

The first study of pregnant women in Asturias was carried out in 2014, when it was found that the median UIC was 197 μg/L [28]. The study highlighted the fact that women who consumed iodised salt but do not take iodine supplements had a median UIC of 190 μg/L, thereby confirming the notion that appropriate nutritional iodine status can be achieved without pharmacological supplementation. Owing to these results, in January 2015 a technical report from the Regional Health Department of Asturias was published that recommended that prescribing iodine supplements for pregnant or breastfeeding women should be made on an individualised basis [29]. The use of supplements for women habitually using iodised salt and consuming 3 servings of dairy products a day was not recommended.

The objective of this study was to ascertain the nutritional iodine status of pregnant women in our health area. In addition, the influence of the ingestion of iodised salt and dairy products as well as the taking of iodine supplements on ioduria and thyroid function were also analysed. Finally, the degree to which the new recommendations from the Regional Health Department have been implemented was assessed.

## 2. Materials and Methods

### 2.1. Study Population

We conducted a descriptive and analytical observational prospective study of women in the first trimester of pregnancy. Recruitment to the study was carried out among all women attending the first of their antenatal visits to the midwife in the Health Centre in Health Area IV in Asturias between May and June 2017.

The following exclusion criteria were applied: multiple gestation, diagnosis of thyroid illness before the pregnancy, treatment within the last three months with thyroid hormone or any product containing high iodine levels.

The study was approved by the Clinical Studies Ethics Committee of Principado de Asturias. All participants signed an informed consent form.

### 2.2. Iodine Consumption Questionnaire

On their first visit to the midwife, participants completed a questionnaire to evaluate their ingestion of iodine. The questionnaire asked about the following:Habitual consumption of iodised salt (yes/no/not known).Consumption of dairy products: number of glasses of milk (1 glass = 200 mL), yoghurts consumed per day and daily consumption of cheese (yes/no). One serving of dairy product was taken as a glass of milk, 2 yoghurts or 80 g of soft or 40 g of hard cheese.Consumption of iodine supplements (yes/no) and date when iodine supplementation began.

Sufficient nutritional ingestion of iodine was considered to be the habitual use of iodised salt and ≥3 servings of dairy products consumed daily.

### 2.3. Urinary Iodine Concentration and Thyroid Function

In order to establish nutritional iodine status, random urine samples were taken. The UIC was determined via inductively coupled plasma-mass spectrometry (ICP-MS) using an ICP-MS 7700x instrument (from Agilent Technologies, Santa Clara, CA, USA). The method showed good linearity between 10 and 450 μg/L (R2 > 0.99), with an intra-laboratory imprecision of ≤2.9% and a total error of ≤7.3%.

At the same time, a blood sample was taken to determine levels of thyroid stimulating hormone (TSH), free thyroxine (FT4) and thyroid autoimmunity (antiperoxidase antibodies, TPOAb, and antithyroglobulin antibodies, TgAb). The analyses were made through electrochemiluminescence immunoassay (Roche Diagnostics, Basel, Switzerland). These determined the levels of TSH (reference range (RR)) in the first trimester in the Oviedo area as: 0.20–4.5 mIU/L and coefficient of variance (CV): 0.8–2.9%; FT4 (RR: 0.99–1.86 ng/dL and CV: 1.8–3.2%); TPOAb (RR: < 34 UI/mL); and TgAb (RR: < 18 UI/mL).

### 2.4. Statistical Analysis

A descriptive analysis was performed to ascertain the distribution and relative and absolute frequencies for qualitative variables, and position and dispersion measurements for the quantitative variables (average and standard deviation (SD) and median and interquartile range (IQR) according to normality criteria).

The differences in numerical variables between two groups were evaluated through Student t-tests. The relationships between qualitative variables were analysed using a Pearson’s chi-squared test. Two multivariate binary logistic regression models were used to assess the factors that could influence the UIC levels below 150 μg/L.

The data from this work are registered at ACCESS-SQL 2010. The statistical analysis was carried out using the R program (R Development Core Team), version 3.6.0. A significance level of 0.05 was used throughout.

## 3. Results

Data was collected from 332 pregnant women but data from 14 subjects was discounted because it met one or more of the exclusion criteria. As such, final analyses were based on data from 318 pregnant women with an average age of 34.10 ± 5.45 years. The average gestation age at the time of the data collection was 7.19 ± 2 weeks.

### 3.1. Iodine Consumption Questionnaire

The iodine consumption questionnaire was completed correctly by 274 women. Data on the use of iodine supplementation was available for 300 women through the questionnaire responses and clinical records.

A total of 51.10% (*n* = 140) of the women habitually used iodised salt, although 7.66% did not know what type of salt they consumed.

In terms of the consumption of dairy products, 48.90% (*n* = 134) of the women consumed two or more servings a day, while 21.17% (*n* = 58) consumed 3 or more. Average consumption was 1.88 ± 1.13 servings a day.

Finally, 87.08% (*n* = 263) of the pregnant women studied took iodine supplements, with 14.57% having started in the preconception period and 78.12% starting before week 13 of their pregnancy. The average dose of iodine supplementation was 206.80 ± 24.15 μg/day.

Combining the data, 29.08% of the women consumed both iodised salt and two or more servings of dairy products and only 10.80% (*n* = 27) met the recommended nutritional iodine ingestion target.

No age differences were observed in relation to the consumption of iodised salt, dairy products or the taking of iodine supplements.

### 3.2. Urinary Iodine Concentration

Data on UIC was available for 316 of the pregnant women in the study. Median UIC was 171.5 μg/L (116–265 μg/L), with 60.41% (*n* = 191) having a UIC of 150 μg/L or higher. The distribution of ioduria is shown in Figure 1.

The consumption of iodised salt was significantly related to higher UIC (*p* = 0.016), while there were no differences in terms of the consumption of dairy products (Table 1). The use of iodine supplements was also significantly related to higher ioduria (*p* = 0.027) (Table 1) although there were no differences in relation to when the women began to take supplements (*p* = 0.279).

The relationship between iodine sufficiency or insufficiency (respectively, ≥150 μg/L and <150 μg/L) was analysed with respect to the questionnaire data. The consumption of iodised salt was associated with UIC ≥ 150 μg/L (*p* = 0.001), although no association was found between UIC and either consumption of dairy products or the taking of iodine supplements (Table 1).

No differences in ioduria were found between pregnant women with sufficient nutritional ingestion of iodine and those who did not meet the recommended levels (median UIC of 168 μg/L and 173.5 μg/L respectively). Women who had sufficient nutritional iodine ingestion but did not take supplements (*n* = 12) had an average UIC of 153 μg/L (119–173.25 μg/L).

Finally, two multivariate binary logistic regression models were applied using UIC < 150 μg/L as the dependent variable and variables from the questionnaire related to iodine ingestion as the independent variables (Table 2). In model 1, the consumption of iodised salt had a certain protective effect in terms of iodine insufficiency (odds ratio (OR) 0.424 (0.251–0.710), *p* = 0.001), whereas neither dairy product consumption nor iodine supplementation showed any preventative effect. In model 2, once again the consumption of iodised salt demonstrated a protective affect (OR 0.404 [0.237–0.683], *p* = 0.001) while drinking milk approached statistical significance (OR 0.519 [0.266–1.007], *p* = 0.053). As in model 1, no effect was found for iodine supplements.

### 3.3. Thyroid Function

Thyroid function was determined for 316 women and thyroid autoimmunity was analysed in 152 cases.

For the analysis of average TSH, 39 patients were excluded due to them presenting positive thyroid autoimmunity. In addition, those patients with TSH levels compatible with either hypothyroidism or hyperthyroidism were also excluded, irrespective of their thyroid autoimmunity status. The total number of patients analysed was therefore 96. Average TSH was 2.26 ± 0.94 mIU/L and average FT4 was 1.17 ± 0.13 ng/dL.

The relationship between iodine ingestion and thyroid function is shown in Figure 2. No statistically significant relationship was found between thyroid function and consumption of either iodised salt or dairy products. Women taking iodine supplements did, however, show lower TSH levels than those who did not (2.17 ± 0.92 vs. 2.77 ± 1.01 mIU/L), values which approached significance (*p* = 0.059). Of the women taking iodine supplements, 68.40% had levels of TSH < 2.5 mIU/L compared to only 30.00% of those who did not take supplements (*p* = 0.031) (Table 3). No link was observed between TSH level and the consumption of either iodised salt or dairy products, and neither were differences in FT4 found with respect to iodine ingestion.

No relationship was found between UIC and thyroid function. The highest values of TSH (average TSH being 3.44 mIU/L) were, however, found in the group of women with UIC > 500 μg/L, although this group comprised only four women.

## 4. Discussion

The iodine nutrition of pregnant women in Health Area IV of Asturias can be seen to be sufficient. However, if the data is compared to that of a study carried out on the same population in 2013 [28], it can be seen that the average UIC has dropped (from 197 to 171.5 μg/L). This situation is similar, though on a smaller scale, to what has been observed in other countries, where in recent years pregnant women have gone from being iodine sufficient to presenting with insufficient iodine states [30,31,32].

The principal measure taken to eradicate iodine deficiency is the universal addition of iodine to salt. In our study it was observed that the use of iodised salt has increased from 46.82% [28] to 51.10%, although this remains considerably less than the 69.30% found in the same region by other authors in 2010 [33]. This percentage of iodised salt consumption in Asturian homes, as has been documented in other regions [22,23,24,34,35], is far below the target of over 90% set by the WHO and the International Council for the Control of Iodine Deficiency Disorders [36]. However, it is important to take into account that the consumption of iodised salt usually begins before the onset of pregnancy; in Spain it has been shown that more than 85% of pregnant women who consume iodised salt have been doing so for at least a year before becoming pregnant [35].

A second and unexpected source of iodine in the diet is through products of animal origin. The addition of iodine to animal feed and the use of iodine supplements in poultry and livestock rearing, as well as the use of products with a high iodine content to clean dairy cows’ udders have resulted in increased iodine levels in foodstuffs of animal origin, especially dairy products and eggs. The incidental contribution of iodine through these foodstuffs translates into an increase in the amount ingested and an improvement in people’s iodine levels [37,38,39]. Indeed, this has been one of the principal contributions to correcting iodine deficiency in recent years and is considered an “accidental triumph for public health” [40,41].

Pregnant women are recommended to consume 2–3 servings of dairy products a day in order to meet their daily iodine requirements [29,42]; one glass of milk is considered to provide 50 μg of iodine, the equivalent of 20% of the recommended intake for pregnant women [43]. In the study Di@bet.es [44] carried out at the national level in Spain between 2009 and 2010, 67.9% of the general population were found to consume two or more servings of dairy products a day, which increased to 79% in the northern part of the country. However, various authors have highlighted a change in eating habits with respect to dairy products, with a marked decline in their consumption [45,46,47]. Our study corroborates this tendency in that only 48.91% of the pregnant women in the study ate two or more servings of dairy products a day compared to 70% in 2013 [28] or 79% in a study of the school-age population in 2010 [27]. The same reduction has also been observed in other studies carried out at the national level [22,23].

To guarantee optimal iodine nutrition during pregnancy, pharmacological supplements are recommended [18,19,20]. In our study, 87.07% of the pregnant women involved were taking supplements (compared to 47% in 2013 [28]). The situation in other regions of Spain is very variable, as was demonstrated in the INMA study conducted between 2003–2008 [48], which described a 97.7% rate of supplementation in Guipúzcoa compared to only 9.4% in Sabadell, although it is also true that the use of iodine supplements has increased across the whole of Spain in recent years [23,24,35].

In our study, both iodised salt and iodine supplements were associated with significant increases in iodine levels. However, in the logistic regression analysis, only iodised salt demonstrated a protective effect against iodine insufficiency regardless of iodine supplementation and consumption of dairy products, which also approached statistical significance. This would explain why despite an increase in the ingestion of iodine supplements in our region in recent years, median UIC has decreased: the drop in the quantity of dairy products being consumed could well be responsible for the reduced iodine levels seen in our area, as is the case in other countries [45,46,47].

Maternal hypothyroidism, whether or not it is the result of iodine deficiency, is linked to increased risk of obstetric complications [49,50] as well as with neurological development disorders in the child [51,52]. For this reason, in spite of principal Endocrinology societies around the world not recommending the universal screening of thyroid function in pregnant women [18,19], the Spanish Society of Endocrinology and Nutrition (SEEN) do advise that it is carried out [53]. The pregnant women in our study had an average TSH level of 2.26 mIU/L, a figure similar to that obtained in 2013 (average TSH of 2.13 mIU/L) [28], although both are higher than results in other recent studies carried out in Spain [24,54]. In a study in Huelva (Andalusia, southern Spain) [54], average TSH in the first trimester was found to be 1.67 mIU/L (NV: 0.246–4.678 mIU/L). Although in this study the iodine status of the pregnant women was not established, we have been able to access data published in one of the Health Areas in that province [55], which describes iodine insufficiency in the population of pregnant women. The situation was similar in a study carried out in Valencia (east coast of Spain) [24], where average TSH was 1.90 mIU/L (NV: 0.128–4.455 mIU/L) in a population of pregnant women with iodine deficiency. One explanation for these differences in our results may lie in the gestational age at the time when samples were taken. There is a drop in TSH levels in the first trimester due to the thyrotrophic effect of chorionic gonadotropin (HCG). This decrease is strongest in week 8 of gestation, coinciding with the peak of HCG [56]. In our study, gestational age was between 4 and 13 weeks, the average being 7.19 weeks. In the Huelva and Valencia studies, simples were taken before the 12th week of gestation, although no average gestational age was reported. It is possible that in both these cases, the average gestational age could be higher than in our study, which would explain the differences in TSH levels.

The effect of iodine supplements on maternal thyroid function and its repercussions on the neurodevelopment of the child in areas of mild to moderate iodine deficiency is a controversial topic. There is little evidence available since there have been few controlled studies and the results of available studies, many of which are observational, are contradictory. In our study those pregnant women who were taking iodine supplements were more likely to have TSH < 2.5 mIU/L than those who were not. These results support the hypothesis that iodine supplementation has a beneficial effect, as has already been demonstrated in other publications [57,58,59]. However, other studies have found supplements to have a neutral effect on maternal thyroid function [60,61,62,63,64] or even to be linked to increases in TSH [28,65,66]. With respect to the effects of iodine supplementation on the foetus, some non-randomised intervention studies have suggested links with improvements in the neurological development of the child [58,67,68]. A review by Taylor [69] found a slight beneficial effect in terms of the cognitive functioning of schoolchildren who were given iodine supplements, although he also explains that the impact of maternal supplements on the neurodevelopment of the foetus remains unknown due to the lack of controlled intervention studies. In the most recent studies, not only have improvements in cognitive functioning not been observed [7,64,66,70,71,72], but a relationship between the use of iodine supplements and worse scores on psychomotor development scales has been identified [65,73]. These results could perhaps be explained by the temporary shock to the thyroid gland as a result of the increase in iodine levels when treatment with supplements begins [74], for which reason it is recommended to start supplementation prior to conception. Two systematic reviews of randomised control trials both concluded that there was no evidence of any beneficial effect on maternal thyroid function or cognitive functioning of the child related to the use of iodine supplements, including when supplementation started preconception or periconception [75,76]. As a result, currently there is no evidence that the use of iodine supplements confers any clinical benefit in areas with mild to moderate levels of iodine deficiency [21,77,78].

In Asturias, since 2014, pharmacological iodine supplementation has been prescribed on an individual basis according to nutritional iodine intake, although this measure has not yet had the hoped-for impact since only 10.8% of pregnant women (*n* = 27) in this study had an adequate nutritional ingestion of iodine. Of these 27 women, only 12 were not taking iodine supplements, although this group did have appropriate median levels of UIC. The sample is too small to draw firm conclusions, but these results go some way to supporting the continuance of the current recommendations of the Health Department in our region.

The two main obstacles to implementing these recommendations are: the low levels of consumption of iodised salt and dairy products as well as the variability in the amount of iodine present in dairy products, which depends on multiple factors such as their geographic origin, how they are processed and the season of the year in which they are produced [79,80]. These problems can be overcome by the universal addition of iodine to salt, campaigns promoting the consumption of dairy products and controls on the quantities of iodine being given to livestock. These measures require political commitment to Public Health Programmes aimed at combating iodine deficiency as well as changes to existing legislation, such as the periodic testing of iodine levels in the general population and in at-risk populations, in addition to collaboration between scientific organisations and national, regional and local government institutions as was proposed by SEEN [81].

The principal limitation of our study is related to the questionnaire about iodine ingestion due to the high number of unusable responses (13.8%) and because the data on the consumption of iodine and dairy products was provided by the pregnant women themselves. In addition, consumption of other foodstuffs such as eggs and fish could have been added to the questionnaire.

## 5. Conclusions

Pregnant women in our Health Area have an adequate nutritional iodine status. The consumption of iodised salt is an effective preventative measure against iodine deficiency regardless of iodine supplementation and does not affect maternal thyroid function. The consumption of dairy products also contributes to obtaining optimal nutritional iodine status, but we need to promote their increased consumption, including in preconception medical consultations. The use of iodine supplements, despite being linked to higher iodine levels, has not, in our study, been shown to be responsible for iodine sufficiency in the pregnant women studied. Given the results obtained here, the policy of prescribing pharmacological iodine supplements on the basis of individual responses to a dietary questionnaire will continue.

## Figures and Tables

**Figure 1 nutrients-13-01816-f001:**
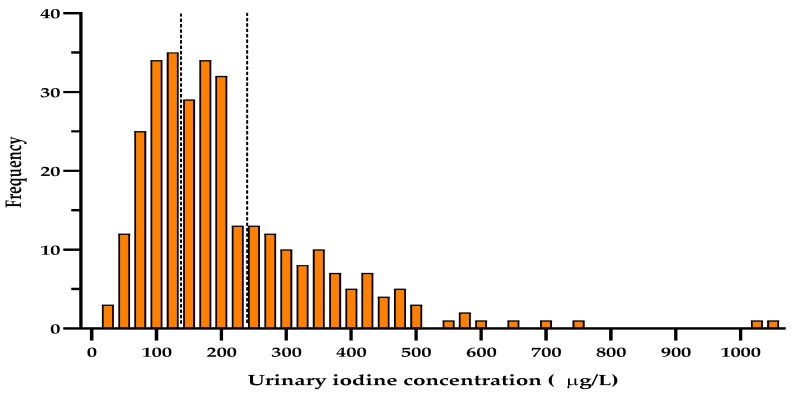
Histogram showing number of woman with different levels of urinary concentration of iodine (μg/L). The dotted lines indicate the recommended range of adequate urinary iodine concentration in pregnant women.

**Figure 2 nutrients-13-01816-f002:**
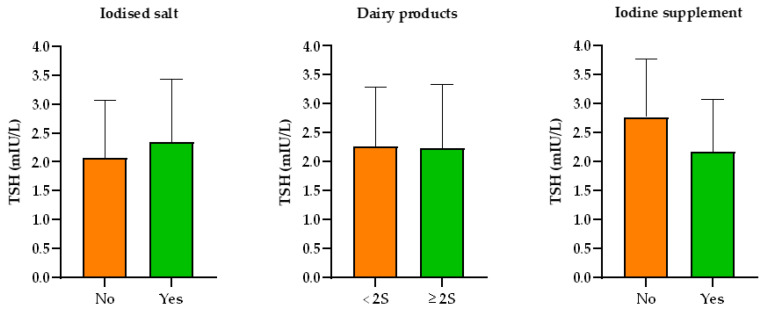
Source of iodine consumption in relation to thyroid function data, the latter is expressed as mean (coloured bars) and SD (whiskers). No statistically significant relationship was found in any scenario. <2S: less than two servings a day; ≥2S: two or more servings a day.

**Table 1 nutrients-13-01816-t001:** Results of iodine consumption questionnaire showing number of women with iodine sufficiency (UIC ≥ 150 μg/L) or insufficiency (UIC < 150 μg/L) as well as median urinary iodine concentration for each group.

		UIC ≥ 150 μg/L	UIC < 150 μg/L	*p*	UIC (μg/L) ^a^	*p*
Iodised salt	No	55 (48.7%)	58 (51.3%)	0.001	147 (102–206)	0.016
	Yes	97 (69.3%)	43 (30.7%)		191.5 (131.5–285)	
Dairy Products ^b^	<2 servings	78 (56.1%)	61 (43.9%)	0.269	168 (96–258)	0.48
	≥2 servings	84 (62.7%)	50 (37.3%)		172 (122.25–255.75)	
Iodine supplements	No	21 (53.9%)	18 (46.1%)	0.455	158.5 (113–199.5)	0.027
	Yes	157 (60.1%)	104 (39.9%)		172.5 (116–285.75)	

^a^ UIC expressed as median and IQR. ^b^ Servings/day.

**Table 2 nutrients-13-01816-t002:** Multivariate binary logistic regression analysis to determine the effect of the ingestion of iodine as protection against iodine insufficiency (UIC < 150 μg/L).

		Β	OR	95% CI ^d^	*p*
Model 1					
Iodised salt	No	Ref.			
	Yes	−0.857	0.424	0.251–0.710	0.001
Dairy products	<2 serving	Ref.			
	≥2 serving	−0.264	0.768	0.439–1.337	0.353
Iodine supplement	No	Ref.			
	Yes	−0.576	0.562	0.259–1.217	0.142
Model 2					
Iodised salt	No	Ref.			
	Yes	−0.905	0.404	0.237–0.683	0.001
Glasses of milk ^a^	0 servings	Ref.			
	≥1 servings	−0.655	0.519	0.266–1.007	0.053
Yoghurts ^b^	0 yoghurts	Ref.			
	≥1 yoghurts	−0.193	0.824	0.467–1.461	0.506
Cheese ^c^	No	Ref.			
	Yes	−0.257	0.784	0.404–1.486	0.462
Iodine supplement	No	Ref.			
	Yes	−0.670	0.512	0.240–1.085	0.080

Results of multivariate binary logistic regression analysis. The dependent variable was UIC < 150 μg/L. ^a^ glasses/day. ^b^ yoghurts/day. ^c^ daily consumption of cheese. ^d^ confidence interval.

**Table 3 nutrients-13-01816-t003:** Iodine consumption data and TSH values.

		TSH ≥ 2.5 mIU/L	TSH < 2.5 mIU/L	*p*
Iodised salt	No	10 (26.3%)	28 (73.7%)	0.13
	Yes	19 (42.2%)	26 (57.8%)	
Dairy products	<2 servings	16 (35.6%)	29 (64.4%)	0.806
	≥2 servings	16 (38.1%)	26 (61.9%)	
Iodine supplement	No	7 (70.0%)	3 (30.0%)	0.031
	Yes	25 (31.6%)	54 (68.4%)	

## Data Availability

Data available on reasonable request from the authors.

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
