# Peer review of "Nutritional Iodine Status in Pregnant Women from Health Area IV in Asturias (Spain): Iodised Salt Is Enough"

_nutrients, 2021, doi:10.3390/nu13061816_

Round 1

Reviewer 1 Report

The main objective of the study “  Appropriate nutritional iodine status in pregnant women from 2 Health Area IV in Asturias (Spain). Is iodised salt enough?  By  Silvia González  Martínez et al,  was , as the authors state,  to ascertain the nutritional iodine status of pregnant  women in their  health area.

Major comments

There is a serious discrepancy between the title and the conclusion in respect with the material and the methods used in the study.

Specifically, both in the title and the conclusions the nutritional status of the women is mentioned, while the vast majority of the subjects were taking supplements  containing unknown amount of iodine.

I suggest the authors should reconstruct the article. Both the title and  especially the discussion and conclusions taking into consideration this fact.

Author Response

Dear reviewer.

First of all I would like to thank you for your comments.

Secondly, before explaining the changes made, I would like to explain the reason for the conclusions drawn. In the multivariate logistic analysis, the influence of each of the variables (iodised salt consumption, dairy products and iodised supplements) on UIC was studied independently. The results of this analysis show that only the consumption of iodised salt has a protective effect on insufficient iodine levels and this effect is independent of the consumption of dairy products and iodised supplements. Therefore, the main conclusion of the study is that iodised salt consumption is sufficient to achieve adequate nutritional iodine status.

Finally, the changes made to the manuscript are listed below:

  • We have modified the title to make it more appropriate.
  • In the results section we have modified the tables at the request of the other reviewer. The average dose of iodine from the supplement has been added.
  • In the discussion the results obtained in the multivariate logistic regression analysis have been explained more explicitly.
  • The main conclusion of the study has been redrafted. This change has also been made to the abstract.

I remain at your disposal should any future changes be necessary.

Reviewer 2 Report

The work is correctly written, but there are big gaps in statistics. When the work is based on multidimensional models, the co-author of the work should be a statistician to ensure correct results and their correct interpretation.

1. Table 1 and Table 3 are hardly legible. I suggest to put the names of the variables in the first column so that we know what "no" and "yes" refer to.
2. In the method section it is stated that the group comparison was done with the Wilcoxon signed-rank test, however, when comparing independent groups this test cannot be used, so you may suspect that the given results (p values) are wrong. Equivalent to Wilcoxon's test for independent groups Mann-Whitney test.
3. Line 192. A p-value was provided which does not appear in any table. All results should be presented in tables or graphs.
4. Line 221. A p-value is given, but it is not known from what table it was derived.
5. In the methods section it is stated that logistic regression models were built for prediction purposes, which is not supported by the presented results. When we build logistics models for the purpose of prediction, we should make a prediction and describe its quality. It seems that these models are designed to assess the factors that could influence the decreased UIC value, not prediction.
6. With reference to the previous comment. In the description of the results of logistic regression models, it should be stated that iodized salt is a protective factor regardless of iodine supplementation and consumption of dairy products. Such conclusions are the advantage of using multivariate models and such conclusions are missing in this work.
7. In Table 2, "a" below the table is unnecessary. This information is described clearly enough in the header of the table.
8. Table 3 should be made in the same way as Table 2. There is no need to put the results describing FT4 on the graphs, the more that these results are statistically insignificant. Usually, we put something on the charts that is to attract attention, because it is important, and in this case, the information is not significant - I suggest transferring this information to Table 3.
9. All tables should be placed in the results section. There is no need to move them to the discussion section.

Author Response

Dear reviewer.

First of all I would like to thank you for your comments.

I would like to explain that this project is supported by the statistical service of the University of Oviedo, which has carried out the data analysis. If necessary, I can send you the reports although they are in Spanish

I respond to your comments below:

  1. Table 1 and Table 3 are hardly legible. I suggest to put the names of the variables in the first column so that we know what "no" and "yes" refer to.

Tables 1 and 3 have been modified following the reviewer's recommendations to facilitate their reading.

  1. In the method section it is stated that the group comparison was done with the Wilcoxon signed-rank test, however, when comparing independent groups this test cannot be used, so you may suspect that the given results (p values) are wrong. Equivalent to Wilcoxon's test for independent groups Mann-Whitney test.

In the statistical analysis part, as your report reflects, there has been a mistake with the statistical tests. The differences in numerical variables between two groups were evaluated solely through Student t-tests. The Wilcoxon test was used to perform another analysis that was not finally included in the manuscript. These data have been confirmed again with the Statistics Service and if you consider it appropriate I can send you the reports specifying the statistical analysis performed at each point.

  1. Line 192. A p-value was provided which does not appear in any table. All results should be presented in tables or graphs.

The p-value has been removed from line 192.

  1. Line 221. A p-value is given, but it is not known from what table it was derived.

We have considered keeping the p-value of line 221 since it seems relevant to us. We know that this value is not found in any table or graph, but we believe that the manuscript already presented too many figures. We can remove it if you do not think it is appropriate.

  1. In the methods section it is stated that logistic regression models were built for prediction purposes, which is not supported by the presented results. When we build logistics models for the purpose of prediction, we should make a prediction and describe its quality. It seems that these models are designed to the decreased UIC value, not prediction.

We have removed the word "predict" for the suggested expression: "assess the factors that could influence”.

  1. With reference to the previous comment. In the description of the results of logistic regression models, it should be stated that iodized salt is a protective factor regardless of iodine supplementation and consumption of dairy products. Such conclusions are the advantage of using multivariate models and such conclusions are missing in this work.

The comment suggested by the reviewer has been added to facilitate the reading of the results obtained in the multivariate logistic regression analysis. This comment has been made in the discussion and conclusions section and the title has been rewritten to explain it better.

  1. In Table 2, "a" below the table is unnecessary. This information is described clearly enough in the header of the table.

It has been removed.

  1. Table 3 should be made in the same way as Table 2. There is no need to put the results describing FT4 on the graphs, the more that these results are statistically insignificant. Usually, we put something on the charts that is to attract attention, because it is important, and in this case, the information is not significant - I suggest transferring this information to Table 3.

Table 2 has been modified. Graphics related to FT4 have been removed.

  1. All tables should be placed in the results section. There is no need to move them to the discussion section.

The tables have been placed in the results section as suggested.

I remain at your disposal should any future changes be necessary.

Round 2

Reviewer 1 Report

I am afraid that the authors did not understand my point of view. I believe that pne cannot come to any conclusion about the effect of the iodized salt on the study population when more than 80% of the participants where on iodine supplemments, regardless of the statistical manipulation.

Author Response

I understand the reviewer's concern. The ideal would be to perform an intervention versus placebo study or an observational study in a population with a lower intake of iodine supplementation. In our case it is an observational study with a high percentage of women with iodine supplementation, indicated on the basis of dietary iodine intake. This may be a limitation but we have two solid statistical analyses that confirm the same data. On the one hand, the multivariate analysis to which I referred in my previous answer and, on the other hand, the analysis shown in Table 1, which shows a higher UIC increase in the comparison between taking or not taking iodized salt vs. taking or not taking iodized supplements (increase of 44.5μg/L vs. 14 μg/L).  Furthermore, it is also confirmed that pregnant women with optimal iodine nutrition without iodized supplementation have adequate UIC.
Therefore, in view of these results we conclude that the consumption of iodized salt has been shown to be the main differentiating element in obtaining correct iodization and is the only element that alone achieves iodine sufficiency in our pregnant women.